# Experimentally Measured Thermal Masses of Adsorption Heat Exchangers

**Kyle R. Gluesenkamp** [1,*]**, Andrea Frazzica** [2]**, Andreas Velte** [3]**, Steven Metcalf** [4]**, Zhiyao Yang** [1,5]**, Mina Rouhani** [6]**, Corey Blackman** [7,8,9]**, Ming Qu** [5]**, Eric Laurenz** [3]**, Angeles Rivero-Pacho** [4]**, Sam Hinmers** [4]**, Robert Critoph** [4]**, Majid Bahrami** [6]**, Gerrit Füldner** [3] **and Ingemar Hallin** [10]

[1]  Oak Ridge National Laboratory, Oak Ridge, TN 37830, USA
[2]  Consiglio Nazionale delle Ricerche (CNR), Istituto di Tecnologie Avanzate per l'Energia "Nicola Giordano" (ITAE), 98126 Messina, Italy; andrea.frazzica@itae.cnr.it
[3]  Fraunhofer Institute for Solar Energy Systems ISE, 79110 Freiburg, Germany; gerrit.fueldner@ise.fraunhofer.de (G.F.); eric.laurenz@ise.fraunhofer.de (E.L.); andreas.velte@ise.fraunhofer.de (A.V.)
[4]  School of Engineering, University of Warwick, Coventry CV47AL, UK; Steven.Metcalf@warwick.ac.uk (S.M.); A.Rivero-Pacho@warwick.ac.uk (A.R.-P.); S.Hinmers@warwick.ac.uk (S.H.); R.E.Critoph@warwick.ac.uk (R.C.)
[5]  Lyles School of Civil Engineering, Purdue University, West Lafayette, IN 47907, USA; yang573@purdue.edu (Z.Y.); mqu@purdue.edu (M.Q.)
[6]  Laboratory for Alternative Energy Conversion, School of Mechatronic Systems Engineering, Simon Fraser University, Surrey, BC V3T0A3, Canada
[7]  SaltX Technology AB, Västertorpsvägen 135, 12944 Hägersten, Sweden
[8]  School of Technology and Business Studies, Dalarna University, 78170 Borlänge, Sweden
[9]  School of Business, Society & Engineering, Mälardalens University, 72123 Västerås, Sweden
[10]  HeatAmp Sweden AB, 11332 Stockholm, Sweden
*  Correspondence: gluesenkampk@ornl.gov

**Abstract:** The thermal masses of components influence the performance of many adsorption heat pump systems. However, typically when experimental adsorption systems are reported, data on thermal mass are missing or incomplete. This work provides original measurements of the thermal masses for experimental sorption heat exchanger hardware. Much of this hardware was previously reported in the literature, but without detailed thermal mass data. The data reported in this work are the first values reported in the literature to thoroughly account for all thermal masses, including heat transfer fluid. The impact of thermal mass on system performance is also discussed, with detailed calculation left for future work. The degree to which heat transfer fluid contributes to overall effective thermal mass is also discussed, with detailed calculation left for future work. This work provides a framework for future reporting of experimental thermal masses. The utilization of this framework will enrich the data available for model validation and provide a more thorough accounting of adsorption heat pumps.

**Keywords:** adsorption; thermal mass; mass ratio; inactive mass; specific thermal mass; resorption

## 1. Introduction

Many adsorption systems (including chemisorption systems) are operated with intermittent heating and cooling of components. Thus, the thermal mass (*TM*; the product of mass and specific heat capacity in units of kJ/K) of the components influences performance. For example, Ziegler (2002) [1] concluded from an overview of existing studies that a key challenge for adsorption systems is the

temperature cycling of *TM* that negatively affects efficiency. Wittstadt et al. (2017) [2] reviewed the recent development for adsorption heat exchangers and pointed out that *TM* affects efficiency and power density within the cyclic operation of adsorption heat pumps. Furthermore, modeling of such systems requires an accounting of the *TM*. However, limited experimental data are available that provide a full accounting of the full *TM* of sorption components.

An inherent tradeoff exists between the thermal cooling or heating capacity of a system and its *TM*. The most efficient cyclic adsorption heat exchanger (HX) will tend toward zero *TM*, but at the expense of reduced heat transfer surface area, thereby leading to longer cycle times, larger size, and lower power density than a less efficient design with higher *TM*. Metcalf and Critoph (2004) [3] investigated the heat and mass transfer intensification limits for carbon—ammonia heat pumps but found that the problem is largely in what can be manufactured.

The link between *TM* and performance has been discussed in the literature. Goetz et al. (1993) [4] studied the effect of the *TM* of the reactive salts and the exchanger and suggested that, based on the experimental results from Neveu (1990) and Douss (1988) [5,6], the drop in the coefficient of performance (COP) due to the *TM* was expected to be around 5–10%.

Li et al. (2009) [7] investigated the impact of the mass ratio between the metallic part of the reactor and the reactive salt in thermochemical refrigeration systems using a thermodynamic model. The mass ratio for an optimally designed solid–gas sorption system was estimated to be around 5:1. Under a higher mass ratio, the added reactor *TM* increased both desorption heat consumption and adsorption heat production without affecting the cold production, which reduced the cooling COP.

Demir et al. (2008) [8] identified *TM* as a critical parameter for cycle times. Paul et al. (2018) [9] studied how to improve efficiency by reducing *TM* by implementing microchannel HXs.

There are several terms known for *TM* in the literature and the research community; Table 1 summarizes the common terms. Heat exchanger materials are commonly divided into two categories, one representing the sorbent material itself and the other representing all other materials, with analogy to a colloquially familiar dichotomy such as living/dead, host/guest, or simply active/inactive. Identifying the active (adsorbent) mass and inactive (non-adsorbent) mass is relatively straightforward. However, identifying the inactive *TM* is more complicated. Among the terms for *TM*, perhaps the primary one is "dead thermal mass." Little was found in the literature for ratios of *TMs*, although "dead thermal mass ratio" was used by Gluesenkamp (2012) and Zhu et al. (2019) [10,11].

**Table 1.** Common terms relating to adsorption heat exchanger thermal mass.

| Sorbent Material Mass (kg) | Non-Sorbent Material Mass (kg) |
|---|---|
| Live mass | Dead mass |
| Active mass | Inactive mass |
| Active mass | Inert mass |
| Host or active mass | Guest material |

The sorbent material itself has *TM*. Thus, while it is logical to distinguish between live and dead mass, a distinction between live and dead *TM* is not very useful. For this reason, this paper uses "thermal mass" and does not use "dead thermal mass."

There are three metrics utilized in this paper: mass ratio, specific *TM* (*STM*), and effective specific heat, as defined in detail in Section 2. Only mass ratio is commonly reported in existing literature.

In the following sections, mass ratios and *TMs* in the literature are summarized.

## 1.1. Mass Ratios of Heat Exchangers

The mass ratio of adsorber metallic masses to adsorbent material was referred to as the "adsorber bed to adsorbent mass ratio (AAMR)" by Sharafian et al. (2016) [12]. This concept is referred to as simply "mass ratio" in the present work.

The mass ratio of adsorber bed metal to adsorbent mass for 18 fin and tube adsorber bed systems reported in the literature is summarized by Sharafian et al. (2016) [12]. Reported values range from as low as 0.654 to as high as 20.9 kg/kg, with 88% of values being between 1.4 and 7.9 kg/kg. These reported values are strictly mass ratios and do not account for *TM*.

Additional mass ratios reported in the literature are summarized by Sharafian and Bahrami (2014) [13]. This list includes ten finned tube adsorbers that are also reported by Sharafian et al. (2016) [12], plus 16 more adsorbers of other types including plate, plate–fin, shell–tube, hairpin, and annulus tube. The mass ratio values for the non-finned tube types reported by Sharafian and Bahrami (2014) [13] ranged from as low as 2.0 to as high as 13.3 kg/kg.

## 1.2. Thermal Masses of Heat Exchangers

The literature was surveyed for reporting of *TM*, as summarized in Table 2. The authors found that only mass ratio is commonly reported in the existing literature, although some cases were identified from which *TM* can be extracted.

**Table 2.** Studies in the literature that provide all or some data needed to calculate heat exchanger (HX) thermal mass (*TM*). HTS, high-temperature salt; LTS, low-temperature salt.

| System and Working Pair | HX | HTF | Specific Thermal Mass: $TM/m_{sorb}$ (kJ$^1$K$^{-1}$kg$_{sorb}$$^{-1}$) | $c_{effective}$: $TM/m_{HX}$ (kJ$^1$K$^{-1}$kg$_{HX}$$^{-1}$) | Reference |
|---|---|---|---|---|---|
| Adsorption: water/FAM-Z02 | Coated round tube–plain fin | Water | 5.62* | 1.64* | [10,14] |
| Adsorption: water/FAM-Z02 | Coated round tube–corrugated fin | Silicon oil | 4.17* | 0.77* | [15] |
| | | Water | 7.01* | 1.28* | |
| Adsorption: water/FAM-Z02 | Packed round tube–corrugated fin | Silicon oil | 5.96*[1] 2.53*[2] | 0.77*[1] 0.78*[2] | [12,15] |
| | | Water | 10.30*[1] 3.98*[2] | 1.31*[1] 1.21*[2] | |
| Resorption: ammonia/MnCl$_2$ (HTS)/BaCl$_2$ (LTS) | Packed shell–tube | Not reported | 2.34[†] (HTS) 2.18[†] (LTS) | 0.49[†] (HTS) 0.47[†] (LTS) | [18] |
| Resorption: ammonia/MnCl$_2$ (HTS)/NH$_4$Cl (LTS) | Coated annular tube | Not reported | 4.77[†] (HTS) 3.85[†] (LTS) | 0.47[†] (HTS) 0.48[†] (LTS) | [16] |
| Resorption: ammonia/MnCl$_2$ (HTS)/NaBr (LTS) | Packed fin–tube | Not reported | 4.55[†] (HTS) 5.00[†] (LTS) | 0.47[†] (HTS) 0.47[†] (LTS) | [17] |

* Includes HTF; [†] HTF data not provided and cannot be directly compared to other values in this work; [1] $m_{sorb}$ = 0.5 kg; [2] $m_{sorb}$ = 1.5 kg.

Table 2 reports the *TM* of HXs reported in the literature. Two normalizations were calculated: *TM* normalized against the sorbent mass ($TM/m_{sorb}$; specific thermal mass [*STM*]) and *TM* normalized against the total HX mass ($TM/m_{HX}$; effective specific heat [$c_{effective}$]). Each of these terms is described in more detail in Section 2. The numerical values of *STM* and $c_{effective}$ were not reported in the original reference, but were computed in this work based on the geometry reported in the original reference.

Only two experimental systems were found in the literature that reported all of the data required to calculate a full accounting of HX *TM*. From Gluesenkamp (2019) and Qian et al. (2013) [10,14], full geometric details were provided for an adsorption HX adsorber/desorber with water as refrigerant and FAM-Z02 as adsorbent, including heat transfer fluid (HTF). From Sharafian et al. (2016) and Rouhani (2019) [12,15], full geometric details were also provided for a water/FAM-Z02 adsorber, including HTF. The component described by Sharafian et al. (2016) and Rouhani (2019) [12,15] was investigated in both coated and packed configurations. The packed configuration was packed with two quantities of sorbent, 0.5 and 1.5 kg. All three studies (coated, 0.5 kg packed, and 1.5 kg packed) are shown in Table 2.

In three additional studies from Xu et al. (2011), Bao et al. (2011), and Lepinasse et al. (1994) [16–18], a nearly complete dataset has been provided to calculate *TM* for resorption HXs; however, the HTF

data were not provided (Table 2). Had the HTF been included, the reported *STM* would have been higher for these HXs.

This work aims to cover the gap in knowledge regarding adsorption HX *TM* by reporting new experimental data for several pieces of experimental hardware from several research laboratories around the world. This research provides new experimental data to the literature with a full accounting of *TM* for fabricated experimental adsorption components. By describing the experimental data in terms of *STM* and $c_{effective}$, the authors also provide a useful correlation between the easily-measured mass ratio and the difficult-to-measure *STM*. Mass ratio is easily measured in the laboratory but not directly useful to system simulation or performance prediction. *STM* is useful to modeling and performance prediction. This work focuses only on experimental measurements and does not attempt to quantify any correlation between *STM* and system performance.

## 2. Methodology

### 2.1. The Definition of Heat Exchanger Thermal Mass

In general, *TM* is the product of mass and specific heat capacity in units of kJ/K. This work only examines the *TM* of adsorption HXs. In adsorption HXs, both the sorbent (the live mass) and the dead mass have *TM*. The *TM* of live mass is inherent to the sorbent material and is thus called $TM_{inherent}$. The amount of dead mass depends on the HX design, and, thus, the *TM* of non-sorbent materials is called $TM_{design}$. The sum of these *TMs* is the total *TM*, $TM_{total}$, as shown in Equation (1).

$$TM_{total} = TM_{inherent} + TM_{design} \quad \left(\frac{kJ}{K}\right) \tag{1}$$

The $TM_{design}$ can be written as in Equation (2). It is the sum of the heat transfer fluid (HTF) thermal mass and the thermal masses of *N* number of materials of construction ("materials" or "mat"). In this work, binders or other materials integrated with the sorbent material are considered as part of the sorbent (unless otherwise noted), and are thus not explicit in Equation (2). The role of HTF is addressed in detail in the next section. Since many material types can be used in HX construction (e.g., copper, aluminum, polymers, and steel), each with their own properties, the materials are written as a sum of *i* = 1 to *i* = N material types.

$$TM_{design} = \rho_{HTF}V_{HTF}c_{HTF} + \sum_{i=1}^{N}(\rho_{mat,i}V_{mat,i}c_{mat,i}) \left(\frac{kJ}{K}\right) \tag{2}$$

The $TM_{inherent}$ can be written as in Equation (3), where *Y* is the mass ratio refrigerant sorbed in the sorbent ($kg_{ref}/kg_{sorbent}$), which varies over time. The product $m_{sorbent}YC_{p,ref,adsorbed}$ is the thermal mass of the refrigerant sorbed in the sorbent. This thermal mass will be lower during heating (from $T_{adsorption}$ to $T_{desorption}$) than during cooling (from $T_{desorption}$ to $T_{adsorption}$) since less refrigerant is retained in the sorbent after completing desorption.

$$TM_{inherent} = m_{sorbent}\left(c_{sorbent} + Yc_{ref,sorbed}\right)\left(\frac{kJ}{K}\right) \tag{3}$$

Neglecting the sorbed refrigerant would simplify the calculation of $TM_{inherent}$ in two significant ways: (1) the calculated thermal mass no longer involves the equilibrium composition of the sorbent; and (2) the thermal mass can be treated as constant in both sorption and desorption processes. For many adsorption HXs, neglecting the *TM* of sorbed refrigerant will have only a small impact on $TM_{total}$ (even though it may be a significant portion of $TM_{inherent}$). In these cases, Equation (3) simplifies significantly to Equation (4).

$$TM_{inherent} = m_{sorbent}c_{sorbent}\left(\frac{kJ}{K}\right) \tag{4}$$

The approach of neglecting sorbed refrigerant thermal mass (i.e., Equation (4)) is utilized throughout this work. Quantification of the impact of neglecting sorbed refrigerant, and the expected dependence on working pair or HX design, is left for future studies. For the present study, this choice of scope allows the focus to be on providing a dataset of experimental values. It is important to note that this represents a limitation of the present study, and it may not be relevant for working pairs with a large quantity of retained refrigerant, or for very low thermal mass HX designs where $TM_{design}$ is small compared to $TM_{inherent}$ and $m_{sorbent}YC_{p,ref,sorbed}$ is a significant fraction of $TM_{total}$.

To summarize, the definition of $TM_{total}$ used in this work is shown in Equation (5). This definition of $TM_{total}$ does not depend on sorbent composition, under the assumption that the sorbate's thermal mass is neglected. All $TM_{totals}$ were evaluated using the material properties specified in Table 3. Any temperature-dependence of these properties was neglected and the fixed values in Table 3 were used.

$$TM_{total} = \rho_{\text{HTF}}V_{\text{HTF}}c_{\text{HTF}} + \sum_{i=1}^{N}(\rho_{mat,i}V_{mat,i}c_{mat,i}) + m_{sorbent}c_{sorbent} \left(\frac{\text{kJ}}{\text{K}}\right) \tag{5}$$

**Table 3.** Thermophysical data.

| Material | Specific Heat Capacity $(\text{kJ}^1\text{kg}^{-1}\text{K}^{-1})$ | Density $(\text{kg}^1\text{m}^{-3})$ | Reference |
|---|---|---|---|
| Activated carbon (monolithic) | 1.05 | 750 | [19] |
| Aluminum (purity > 99%) | 0.91 | 2700 | [20] |
| Ammonia (liquid) | 4.60 | 639 | [21] |
| Binder SilRes MP50 E | 1.27 | Not used | Measured |
| TiAPSO SCT-323 | 0.90 | Not used | Estimated value |
| Adsorbent SAPO-34 directly crystallized | 0.90 | 1500 | [22] |
| Copper | 0.385 | 8.96 | [23] |
| Water | 4.19 | 1,000 | [20] |
| AQSOA FAM Z02 grains | 0.69 | Not used | [24] |
| AQSOA FAM Z02 powder | 0.822 | 600–700 | [24,25] |
| Silicon oil (HL80, Julabo) | 1.726 | 910 | [26] |
| Siogel | 0.62 | Not used | [24] |
| Activated carbon | 1.1 | Not used | [27] |
| Silane binder | 1.3 | Not used | [28] |
| Stainless steel (316) | 0.49 | 7954 | [29] |
| MnCl$_2$–graphite 1:2 mixture | 1.24 | 470 | [16] |
| NH$_4$Cl–graphite 1:2 mixture | 0.61[1] | 470 | [16] |
| MnCl$_2$–graphite 13:7 mixture | 0.6[1] | 495 | [17,18] |
| BaCl$_2$–graphite 13:7 mixture | 0.5[1] | 507 | [17,18] |
| MnCl$_2$–graphite 13:7 mixture | 0.61[1] | 310 | [17] |
| NaBr–graphite 13:7 mixture | 0.54[1] | 300 | [17] |

[1] Calculated based on weighted average of $c_{binder}$ and $c_{salt}$, and mass ratio of binder/salt provided in the reference.

Additional terms used in this work are expressed in Equations (6)–(9). Equation (6) defines the HX mass ($m_{\text{HX}}$ [kg]) as the sum of sorbent, HTF, and materials of construction. Equation (7) defines the mass ratio (*MR*, [kg/kg]) as the ratio of $m_{\text{HX}}$ to mass of sorbent ($m_{sorb}$). Equation (8) defines specific

thermal mass $STM$ ($kJ^1K^{-1}kg_{sorb}{}^{-1}$) as the $TM_{total}$ per unit sorbent mass. Equation (9) defines the effective specific heat ($c_{effective}$, [$kJ^1K^{-1}kg_{HX}{}^{-1}$]) as the $TM_{total}$ per unit HX mass.

The utility in defining $c_{effective}$ this way is that it provides a path to a straightforward translation between an easily measured quantity (component $MR$) and a less easily measured one (component $TM_{total}$). In other words, if one seeks to know the $TM_{total}$ of an HX for which a detailed measurement of $TM_{total}$ has not been made, the mass of the unknown component can be multiplied by a relevant $c_{effective}$ to obtain an estimated $TM_{total}$.

This translation requires knowledge of typical values of $c_{effective}$. If the $c_{effective}$ is reported for many similar components, a statistical expectation can be established for the typical $c_{effective}$ for that type of component. The work in this paper is the first step to such a catalog of data.

$$m_{HX} = m_{sorb} + m_{HTF} + m_{mat} \ (kg_{HX}) \tag{6}$$

$$MR = \frac{m_{HX}}{m_{sorb}} \left( \frac{kg_{HX}}{kg_{sorbent}} \right) \tag{7}$$

$$STM = \frac{TM}{m_{sorb}} \left( \frac{kJ}{K \cdot kg_{sorbent}} \right) \tag{8}$$

$$c_{effective} = \frac{TM}{m_{HX}} \left( \frac{kJ}{K \cdot kg_{HX}} \right) \tag{9}$$

A useful property of these definitions is that $STM$, $MR$, and $c_{effective}$ are related, as shown in Equation (10). Since $MR$ is much easier to measure than $STM$, Equation (10) can be used to predict $STM$ based on an expected value of $c_{effective}$.

$$STM = c_{effective}MR \left( \frac{kJ}{K \cdot kg_{sorbent}} \right) \tag{10}$$

### 2.2. Definition of a Relevant Control Volume

To apply these definitions to an adsorption HX, a control volume must be chosen. Three possible control volumes are depicted in Figure 1a,c for three classes of HX. The comprehensive control volume has the advantage of including all thermal masses expected to undergo a temperature change with each adsorption/desorption cycle, including HTF and plumbing between the HX and the switching valves, which is relevant to whole-system performance. However, the system is highly dependent on any particular implementation of an HX. In addition, many laboratory/experimental systems are not relevant to full systems because they are small-scale and lack a header or include instrumentation and other components within the comprehensive control volume, which would not be present in a commercial system.

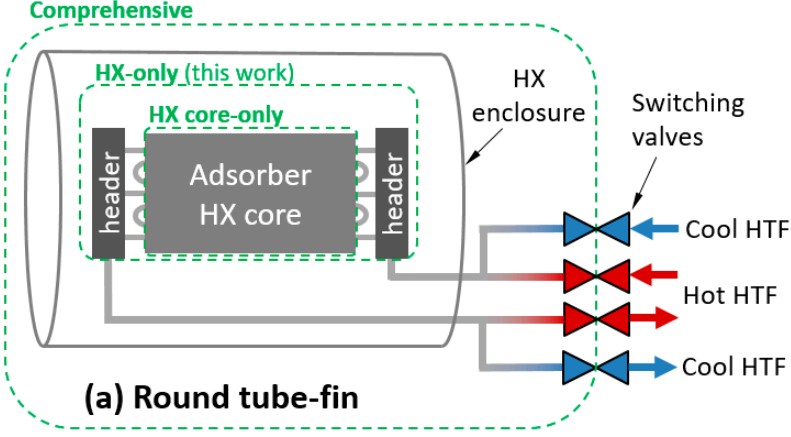

**Figure 1.** *Cont.*

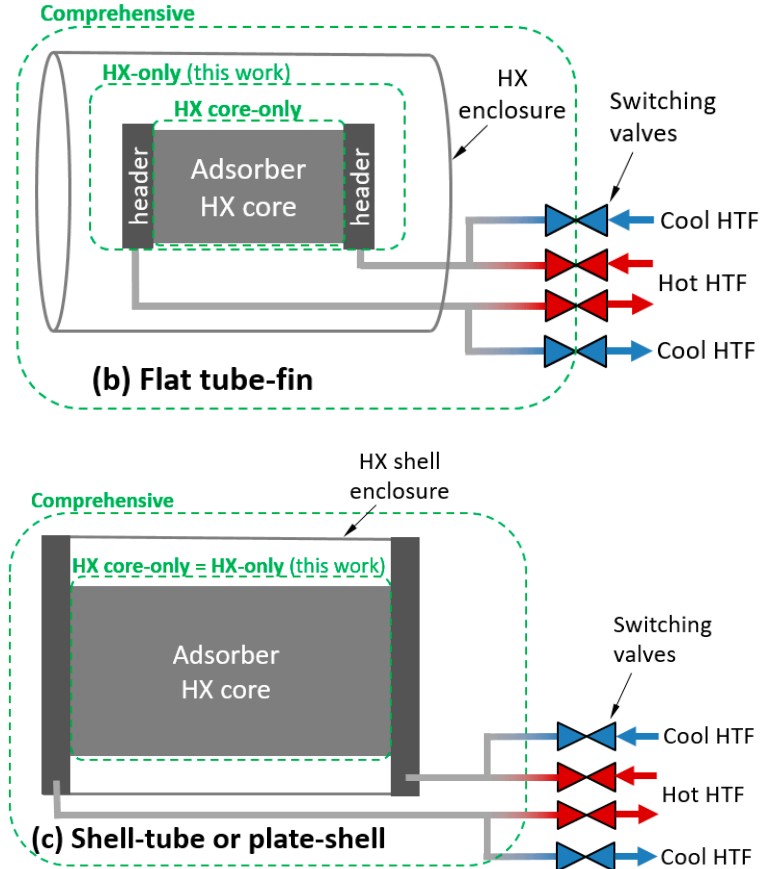

**Figure 1.** Illustration of control volumes and heat transfer fluid (HTF) relevant to thermal mass for: (**a**) round tube–fin; (**b**) flat tube–fin; and (**c**) shell–tube or plate–shell HXs.

Treating the comprehensive control volume is complicated because significant spatial temperature variations can exist within the control volume, and can vary with number of cycles. For example, if the sorbent is cycled between 25 and 100 °C in adsorption and desorption phases, the enclosure might experience minimum and maximum temperatures of 25 and 35 °C in the first cycle, and stabilize to minimum and maximum temperatures of 40 and 60 °C after several cycles. In contrast, the "HX-only" control volume can usually be characterized by a single set of minimum and maximum temperatures.

For simplicity, values in this work are reported for the HX-only control volume. The HX-only control volume includes the sorbent, metal, and HTF within the core and headers. It excludes the enclosure and any plumbing between the headers and the switching valves. In the case of a shell–tube or plate–shell HX, the headers and shell are typically inextricable, and have been excluded from the HX-only control volume.

### 2.3. The Role of Heat Transfer Fluid

An emphasis is placed here on HTF because it is the most commonly neglected parameter in reported data for computing thermal mass.

In typical designs, the HTF is a full participant in the thermal cycling behavior of the component. In other words, all the HTF retained in the HX must be fully heated and cooled with each cycle. There may be system-level options to minimize the impact of the HTF thermal mass, but in all cases, the HTF thermal mass contributes to the heat that must be added and removed from the component with each cycle.

The HTF can be accounted for in various control volumes, as illustrated in Figure 1. In this work, the HX-only control volume was used.

### 2.4. The Role of Heat Exchanger Enclosures

The enclosure (or shell) of an HX does not fully participate in the temperature swings experienced by the sorbent and heat exchange materials inside. Thus, the thermal mass of the shell has reduced importance compared with the HX core. The degree of participation depends on factors including the switching duration, the thermal diffusivity of the shell, the effectiveness of heat transfer between the shell and the core (radiation, convection, and conduction mechanisms), and the degree of insulation on the exterior of the shell.

In this work, this complexity is treated by ignoring the thermal mass of the HX shell. A rigorous treatment would require that shell or enclosure thermal mass be considered.

## 3. Experimental Results for Specific Thermal Mass

In this section, experimentally measured masses (and their corresponding $TM_{totals}$) are presented for several sorption HXs. Flat tube–fin HX are investigated, including packed adsorber beds (Sections 3.1 and 3.2) and coated HXs (Sections 3.3 and 3.4). Round tube–corrugated fins are investigated in Section 3.5 (both coated and packed), a modular finned tube in Section 3.6, shell–tube in Section 3.7, plate–shell in Section 3.8, and a fiber HX in Section 3.9.

### 3.1. Flat Tube–Fin—Packed (Water as Refrigerant)

Two packed HX components are described here: one packed with silica gel, and one with zeolite. Different adsorber configurations based on a defined aluminum flat tube–fin HX, shown in Figure 2, were realized and tested by means of a laboratory-scale test rig (cooling capacity up to 1 kW) described by Frazzica et al. (2016) [30]. The HX was characterized by an aluminum mass of 0.51 kg and an HTF volume of 300 cm$^3$ with an HX core-only volume of approximately 1000 cm$^3$. The overall heat transfer area, comprising the fins, accounted for 0.94 m$^2$. As a packed adsorber bed, it was tested using the zeotype material (i.e., grains of AQSOA-Z02) and later with silica gel (grains of Siogel). The mass of AQSOA-Z02 was 0.26 kg, while the mass of Siogel was 0.31 kg. In both cases, the grain size distribution was 0.6–0.8 mm; thus, the mass difference was related to the different density of the material, as well as the shape of the grains. Being characterized by an irregular shape, Siogel had a higher packing density. The specific heat for the two adsorbent materials was derived from the work by Santori et al. (2013) [24]. In particular, average values of 0.75 and 0.72 kJ/kg K were used for AQSOA-Z02 and Siogel, respectively. The experimental characterization of the adsorbers, carried out in a dedicated test rig described by Sapienza et al. (2011) [30], showed an average cooling power of 0.30 and 0.20 kW for the AQSOA-Z02 and Siogel configurations, respectively. No effect of the inert mass of the vacuum chamber was considered in the performance evaluation.

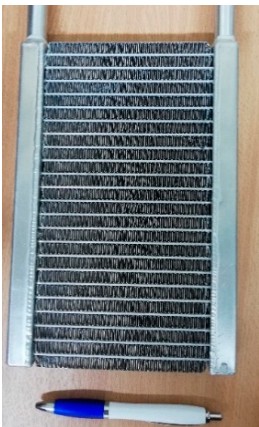

**Figure 2.** Small-scale aluminum heat exchanger [30].

### 3.2. Flat Tube–Fin—Packed (Ethanol as Refrigerant)

A concept similar to that of Section 3.1 was developed for a laboratory-scale activated carbon-ethanol refrigerator, whose nominal cooling capacity was 0.5 kW [31]. For each adsorber, four tube–fin aluminum HXs were employed, such as the one presented in Figure 3, working in parallel. Each adsorption HX had about 1.9 kg of aluminum and hosted 0.6 kg of activated carbon. The internal volume for HTF was 550 cm$^3$. The considered specific heat of the activated carbon was 1.1 kJ/kg K, as reported by Brancato et al. (2015) [27]. In the experimental tests reported by Palomba et al. (2017) [32], the inert mass due to the adsorber vacuum chamber was also considered. Particularly, the thermal cycling of the flanges on top of the chamber and the whole shell was monitored during the testing. The results allowed estimating that the thermal energy input allocated to heating the flanges and shell accounted for about 15% of the total thermal energy used to drive the prototype.

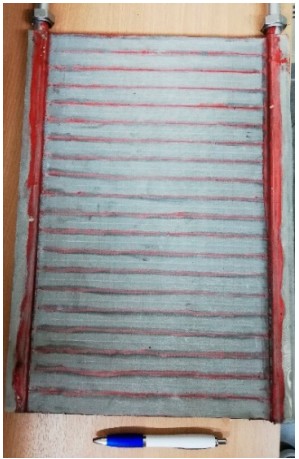

**Figure 3.** Tube–fin aluminum heat exchanger filled with activated carbon for a small-scale adsorption refrigerator, employing ethanol as the refrigerant [31].

### 3.3. Flat Tube–Fin—Coated

An innovative binder-based coating was developed and applied to the HX shown in Figure 4. It employed 90 wt % of AQSOA-Z02 and 10 wt % of silane as the binder, which had a specific heat of 1.3 kJ/kg K [28]. The amount of adsorbent material loaded inside the HX was much lower than that of the packed bed because the coating thickness of 0.12 mm maximizes the heat and mass transfer efficiency, thus achieving a specific power as high as possible.

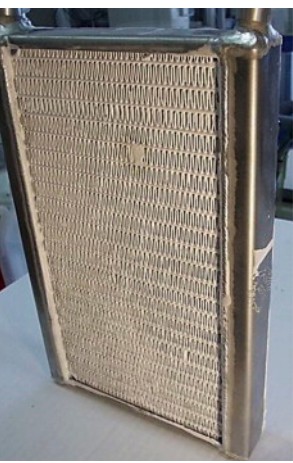

**Figure 4.** Flat tube–fin coated adsorber [28].

### 3.4. Flat Tube–Fin—Coated

Figure 5 shows a series of the flat tube–fin HXs coated with TiAPSO SCT-323 from Clariant AG, Bitterfeld, with binder SILRES MP 50 E from Wacker Chemie AG [33]. The HX without coating and flanges had a weight of 0.465 kg and was completely made of aluminum. The mass of the HTF (water) inside the flat tubes was 0.1 kg, and in the headers and the additional tubes was 0.15 kg.

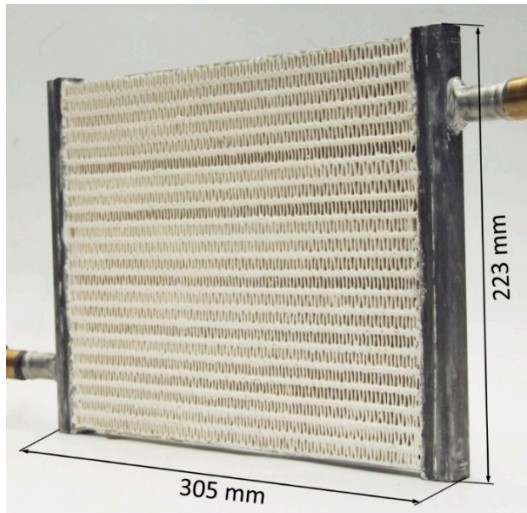

**Figure 5.** Coated flat tube–fin heat exchanger [33].

Five variants with different coating thicknesses were manufactured and measured by Bendix et al. (2017) [33]. The measurements revealed that variant HX-4 with an adsorbent mass of 0.445 kg was a good compromise between efficiency and power density. If the coating thickness (or filling factor) was increased further, heat and mass transfer limitations led to significantly lower power output.

### 3.5. Round Tube–Corrugated Fin—Coated and Packed (Water as Refrigerant)

A custom-built sorption heat pump testbed was designed and used to study the performance of FAM-Z02-coated and -packed adsorber beds [12,15]. As shown in Figure 6, round tube–corrugated fin HXs, manufactured by Hayden Automotive, were chosen as the adsorber beds. The weight of the bare HX ($m_{mat}$) was about 2.6 kg, consisting of 2.1 kg of copper tubes and 0.5 kg of aluminum fins. The internal volume of the HTF was 837 $cm^3$, equivalent to 0.762 kg of silicon oil with density of 910 $kg/m^3$ (at 30 °C) and heat capacity of 1.726 $kJ^1kg^{-1}K^{-1}$ (at 30 °C). The volume of the adsorber bed loaded with sorbent was 4079 $cm^3$, while the total volume of the adsorber bed was 5521 $cm^3$.

In the packed configuration, the HX was filled with 0.5 or 1.5 kg of 2 mm diameter FAM-Z02 particles. In the coated configuration, the identical HX was coated with 0.766 kg of FAM-Z02 in coating thickness of 0.3 mm, fabricated by Mitsubishi Plastics. The overall heat transfer area of the coated bed was 2.8 $m^2$.

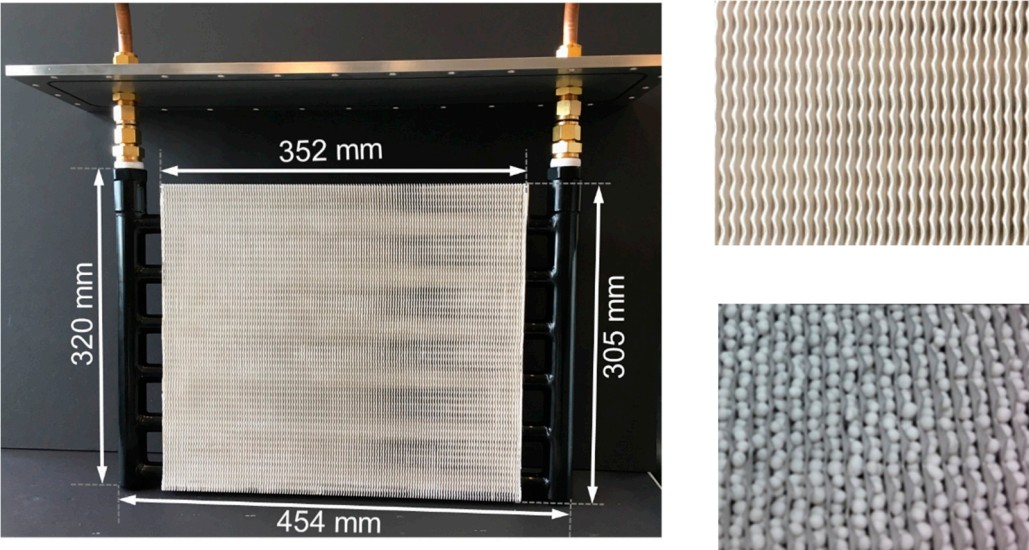

**Figure 6.** Round tube–corrugated fin adsorber bed with FAM-Z02-coated and -packed configurations [12,15].

### 3.6. Modular Finned Tube

Figure 7 shows a modular generator design for a carbon-ammonia heat pump [34]. The modules were heated and cooled by air and were arranged in a rotating bank to allow regeneration approaching counter-flow heat transfer. The thermal mass of the HTF (air) was negligible. However, a significant mass of aluminum fins was required for effective heat transfer. The central 12.7 mm diameter stainless steel tubes contained active carbon and were 600 mm long and had a wall thickness of 0.25 mm and a mass of 93 g. The aluminum fins were 0.3 mm thick on a 1 mm pitch, with a total mass of 543 g. Since the modules rotated, the outer ducting for the air flow was at a constant temperature and thus its thermal mass could be neglected. The highly regenerative cycle allowed for a relatively high COP despite the high thermal mass of aluminum fins, but came at the expense of long cycle time and low power density.

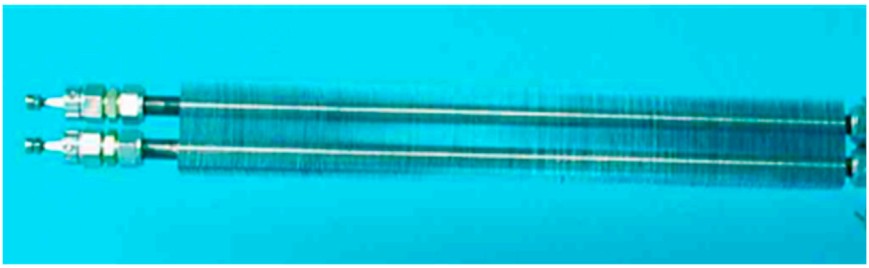

**Figure 7.** Modular finned tube carbon-ammonia adsorber [34].

### 3.7. Shell–Tube

Figure 8 shows the core of a shell–microtube HX developed for a carbon-ammonia adsorption heat pump [35]. The unit was made from nickel-brazed stainless steel, with approximately 800 tubes that were 300 mm long, 1.2 mm in diameter, and 0.2 mm in wall thickness. The unit contained 1.12 kg of active carbon with a density of 750 kg m$^{-3}$. The microtube design gave a good trade-off between short conduction path length in the active carbon for high power density and low $TM_{total}$ for high COP. The HX contained 1.71 kg of stainless steel (1.46 kg in the core and 0.25 kg in the headers) and 0.22 kg of water (0.12 kg in the tubes and 0.1 kg in the headers). The outer shell did not undergo thermal cycling and thus was excluded from the mass of the unit.

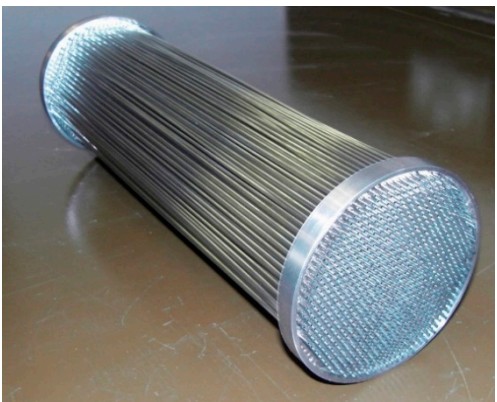

**Figure 8.** Heat exchanger core of ammonia/carbon heat pump generator [35].

## 3.8. Plate–Shell

Figure 9 shows a core of a plate–shell HX for a halide salt-ammonia adsorption heat pump [36]. The HX incorporated a heat pipe, enabling high heat flux and full utilization of the contacting surface area with condensation heat transfer during heating desorption, and convection heat transfer during cooling adsorption.

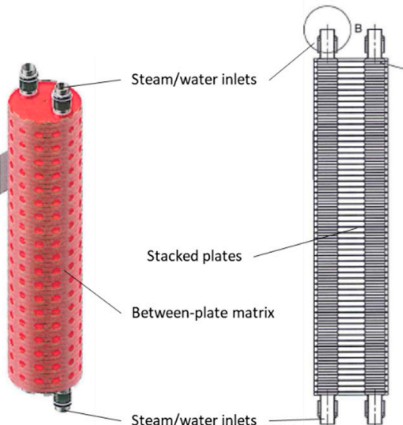

**Figure 9.** Plate–shell heat exchanger.

The unit was made from 56 plates of 316 stainless steel with 3 mm thickness and 10 mm spacing. The HX contained 7.8 kg of stainless steel and 2.36 kg of HTF (water) inside the stacked plates and HX tubes between the inlet and outlet. The unit contained 2.4 kg of nanocoated halide salt and carbon-based matrix materials as the sorbent. The density of the composite sorbent was 520 kg m$^{-3}$. Additionally, the shell of the HX was 24.9 kg of 316 stainless steel. The summary in Table 4 does not include this mass. Experimentally, the shell mass was found to undergo 40% of the temperature change experienced by the halide salt sorbent and active HX surfaces.

**Table 4.** Measured mass ratios (*MR*), specific thermal masses (*STM*), and $c_{effective}$, excluding shells and enclosures for HX-only control volume of adsorber heat exchangers (HXs).

| Component (Section) | Working Pair (refr./Sorbent) | HTF | $TM_{total}$ (kJ$^1$K$^{-1}$) | Sorbent Mass $m_{sorb}$ (kg$_{sorb}$) | HX Mass*$m_{HX}$ (kg$_{HX}$) | MR, $m_{HX}/m_{sorb}$ (kg$_{HX}^1$ kg$_{sorb}^{-1}$) | STM, $TM/m_{sorb}$ (kJ$^1$K$^{-1}$kg$_{sorb}^{-1}$) | $c_{effective}$ $TM/m_{HX}$ (kJ$^1$K$^{-1}$ kg$_{HX}^{-1}$) |
|---|---|---|---|---|---|---|---|---|
| Flat tube–fin, packed (Section 3.1) | Water/silica gel | Water | 1.92 | 0.31 | 1.12 | 3.58 | 6.10 | 1.70 |
| Flat tube–fin, packed (Section 3.1) | Water/zeolite | Water | 1.90 | 0.26 | 1.07 | 4.12 | 7.31 | 1.78 |
| Flat tube–fin, packed (Section 3.2) | Ethanol/AC | Water | 4.66 | 0.60 | 3.05 | 5.08 | 7.77 | 1.53 |
| Flat tube–fin, coated (Section 3.3) | Water/zeolite | Water | 1.80 | 0.092 | 0.90 | 9.80 | 19.58 | 2.00 |
| Flat tube–fin, coated (Section 3.4) | Water/zeolite | Water | 1.97 | 0.45 | 1.24 | 2.77 | 4.40 | 1.59 |
| Round tube–fin, packed (Section 3.5) | Water/FAM-Z02 | Silicon oil | 2.98 | 0.5 | 3.86 | 7.72 | 5.96 | 0.77 |
| | | | 3.80 | 1.5 | 4.86 | 3.24 | 2.53 | 0.78 |
| Round tube–fin, packed (Section 3.5) | Water/FAM-Z02 | Water | 5.15 | 0.5 | 3.94 | 7.87 | 10.30 | 1.31 |
| | | | 5.97 | 1.5 | 4.94 | 3.29 | 3.98 | 1.21 |
| Round tube–fin, coated (Section 3.5) | Water/FAM-Z02 | Silicon oil | 3.20 | 0.766 | 4.12 | 5.39 | 4.17 | 0.77 |
| Round tube–fin, coated (Section 3.5) | Water/FAM-Z02 | Water | 5.37 | 0.766 | 4.20 | 5.49 | 7.01 | 1.28 |
| Round tube–fin, coated (1.2) | Water/zeolite | Water | 14.1 | 2.82 | 9.68 | 1.84 | 5.00 | 1.46 |
| Modular finned tube (Section 3.6) | NH$_3$/A | Water | 0.6 | 0.078 | 0.71 | 9.1 | 7.7 | 0.85 |
| Shell–tube (Section 3.7) | NH$_3$/AC | Water | 2.75 | 1.12 | 3.05 | 2.72 | 2.46 | 0.9 |
| Plate–shell (Section 3.8) | NH$_3$/nanocoated halide salt | Water | 20.8 | 3.4 | 13.56 | 3.99 | 6.12 | 1.53 |
| Flat tube–fiber (Section 3.9) | Water/zeolite | Water | 18.58 | 3.30 | 13.69 | 4.15 | 5.63 | 1.36 |

* Heat exchanger mass includes sorbent mass and HTF mass, per Equation (6). In this table, the HX-only control volume has been used. refr., refrigerant

### 3.9. Fiber Heat Exchangers

Figure 10 shows an HX with flat tubes and fibrous aluminum structures between the flat tubes. The fibrous structures are directly crystallized with SAPO-34 with the partial support transformation (PST) technique [37]. The fibrous structures are made of aluminum fibers sintered together, brazed onto the aluminum flat tubes and finally coated with adsorbent crystals [38]. The first experimental results for directly crystallized fibrous structures showed the potential of this approach to massively increase power density [39]. These findings were confirmed later at the scale of a complete adsorption module including a fiber HX [40]. The fibers had a mean diameter of approximately 130 μm. The mean thickness of the SAPO-34 crystallite layer was around 50 μm. The fibrous structures had a large volume-specific surface area, more than 7000 $m^2/m^3$. The total surface area of the fibers of this HX was approximately 41 $m^2$.

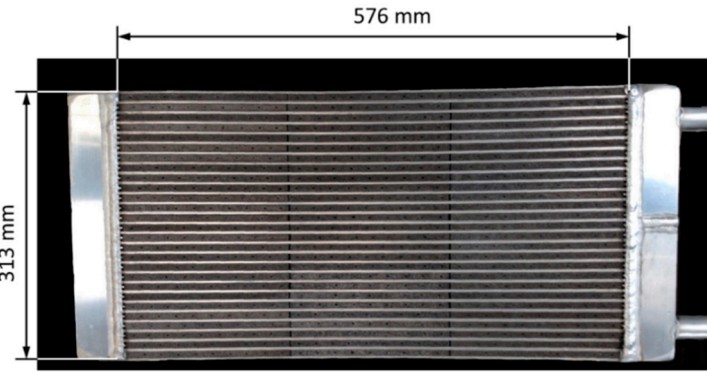

**Figure 10.** Fiber heat exchanger developed and tested by Wittstadt et al. (2017) [40].

The mass of HTF (water) inside the flat tubes was 1.07 kg, and in the header and the additional tubes was 0.83 kg. The measurements of the module indicated that the *TM* of the housing also plays a role since heat is transferred from the adsorber and the combined evaporator–condenser to the housing.

### 3.10. Summary of Experimental Results

The main results of the original measurements made for this paper are summarized in Table 4. For each HX, the $TM_{total}$, sorbent mass, and total heat exchange mass are reported. From these three raw measured values, the *MR*, *STM*, and $c_{effective}$ were calculated according to their definitions in Equations (7)–(9).

Representative propagated measurement uncertainties in $TM_{total}$, *STM*, and $c_{effective}$ were computed for one case. For the plate–shell HX described in Section 3.8, the propagated uncertainties were 2.3% in $TM_{total}$, 2.7% in *STM*, and 2.2% in $c_{effective}$. These were based on measurement uncertainties of 5% in $m_{HTF}$, 2% in $m_{matrix}$ and $m_{sorb}$, and 1% in $m_{HX}$, and property uncertainties of 0.3% in $c_{HTF}$, 1% in $c_{HX}$, and 4% in $c_{matrix}$ and $c_{sorb}$. Other cases are expected to be similar.

Repeatability of measurements was not addressed in this work.

Table 5 provides additional details for each HX studied in this work. This table provides the individual contributions to mass and thermal mass of the three main components: HTF, metal, and sorbent. In most cases, the HTF is the largest contributor to thermal mass.

**Table 5.** Details of measured masses and thermal masses, excluding shells and enclosures for HX-only control volume of adsorber heat exchangers.

| Component (Section) | Masses | | | Thermal Masses | | |
|---|---|---|---|---|---|---|
| | HTF (kg) | Metal (kg) | Sorbent $m_{sorb}$ (kg$_{sorb}$) | HTF (kJ$^1$K$^{-1}$) | Metal (kJ$^1$K$^{-1}$) | Sorbent (kJ$^1$K$^{-1}$) |
| Flat tube–fin, packed (Section 3.1) | 0.3 (water) | 0.51 | 0.31 | 1.25 | 0.46 | 0.226 |
| Flat tube–fin, packed (Section 3.1) | 0.3 (water) | 0.51 | 0.26 | 1.25 | 0.46 | 0.195 |
| Flat tube–fin, packed (Section 3.2) | 0.55 (water) | 1.9 | 0.60 | 2.30 | 1.73 | 0.66 |
| Flat tube–fin, coated (Section 3.3) | 0.3 (water) | 0.51 | 0.084 (sorbent) 0.009 (binder) | 1.25 | 0.46 | 0.074 |
| Flat tube–fin, coated (Section 3.4) | 0.25 (water) | 0.47 (Al) | 0.45 | 1.05 | 0.42 | 0.50[1] |
| Round tube–fin, packed (Section 3.5) | 0.76 (silicon oil) 0.83 (water) | 2.1 (copper) 0.5 (Al) | 0.5 1.5 | 1.31 3.49 | 1.25 | 0.41 1.23 |
| Round tube–fin, coated (Section 3.5) | 0.76 (silicon oil) 0.83 (water) | 2.1 (copper) 0.5 (Al) | 0.766 | 1.31 3.49 | 1.25 | 0.63 |
| Round tube–fin, coated (1.2) | 1.66 (water) | 5.20 | 2.82 | 6.96 | 4.68 | 4.23 |
| Modular finned tube (Section 3.6) | 5e-4 (air) | 0.636 | 0.078 | 5e-4 | 0.53 | 0.07 |
| Shell–tube (Section 3.7) | 0.222 (water) | 1.71 | 1.12 | 0.92 | 0.86 | 1 |
| Plate–shell (Section 3.8) | 2.36 (water) | 7.8 (stainless steel) | 3.4 | 9.86 | 3.82 | 5.44 |
| Fiber heat exchanger (Section 3.9) | 1.9 (water) | 8.5 (Al) | 3.3 | 7.94 | 7.65 | 2.97 |

[1] Includes 0.10 for binder.

## 4. Discussion

The data for the 16 adsorber cases compiled in Table 4 are plotted in Figures 11 and 12. The data are the 15 measurements presented in Section 3, plus one measurement from the literature (described in Section 1.2).

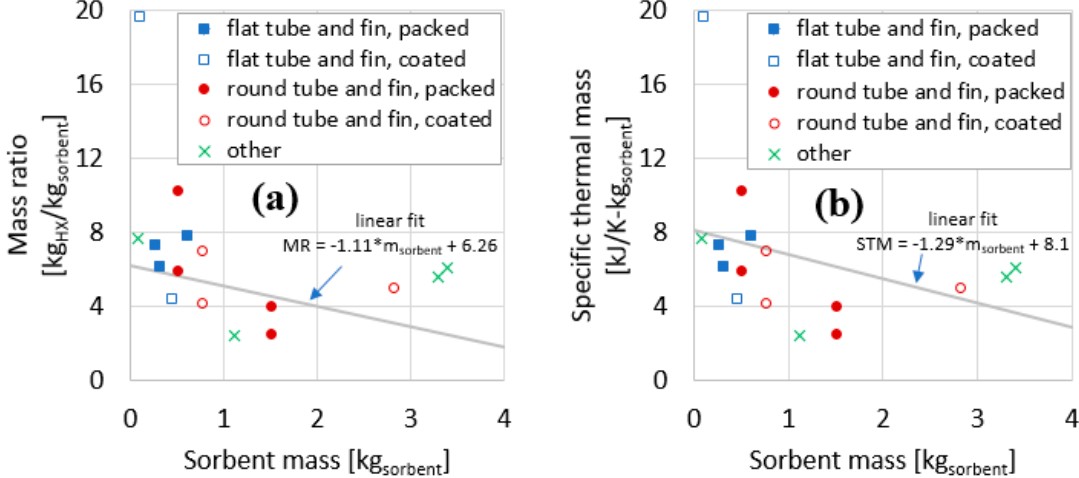

**Figure 11.** Mass ratio (**a**) and specific thermal mass (**b**) plotted against sorbent mass for the components evaluated in this work.

Figure 11a plots the *MR* as a function of sorbent mass. Sorbent mass here is a proxy for HX size. In general, for larger HXs, the *MR* declines. Figure 11b plots the *STM* as a function of sorbent mass. In general, similar to the trend for *MR*, the *STM* declines for larger HXs.

Figure 12 plots *STM* as a function of *MR*. This relationship between *STM* and *MR* has a reduced dependence on HX size. Two linear fit lines are shown: one considering the five flat tube–fin adsorbers (all of which used water as the HTF), and another considering all 13 adsorbers with water as the HTF.

One observation based on Figure 12 is the *STM* can often be predicted fairly well using a simple linear fit to the data. Considering only those adsorbers with water as HTF, $STM = 1.45 \times MR$, where 1.45 is a $c_{effective}$ ($kJ^1K^{-1}kg_{HX}^{-1}$) obtained empirically to fit the data compiled in this work, with the fitted line forced to traverse the origin of the plot (0,0), as required by Equation (10). Specifically, using this fit, the root mean-square error (RMSE) of the prediction relative errors was 32%. The correlations predict the *STM* for 10 of the 13 reported HXs within −28%/+19% maximum relative error. The outliers are the coated round tube–fin from Section 1.2 (−47% prediction error), the modular finned tube from Section 3.6 (+71% prediction error), and the shell–tube from Section 3.7 (+60% prediction error).

Restricting consideration to the flat tube–fin HX (including packed and coated), a separate fit can be determined, in which the $c_{effective} = 1.70$ ($kJ^1K^{-1}kg_{HX}^{-1}$) (in other words, $STM = 1.70 \times MR$). The fit for this geometry is such that all five data points are predicted fit within −15% to +10% prediction error, and the RMSE of the relative errors was 9%.

The three HXs that use silicon oil as the HTF appear as outliers because HTF has a large influence on $TM_{total}$, and changing from high specific heat water (c = 4.19 $kJ^1kg^{-1}K^{-1}$) to low specific heat oil (c = 1.73 $kJ^1kg^{-1}K^{-1}$) has a large influence on *STM*, but a much smaller influence on *MR*.

This study focused on $TM_{total}$ within the HX-only control volume, and Table 6 compares the "HX core-only" to the HX-only control volume for some of the components reported in this study. Data for the HX core-only control volume was only compiled for five of the evaluated adsorbers. By comparing the last two columns in Table 6, it is apparent that significant differences exist between these two control volumes, with the header (also called "collector") contributing an additional 23–63% to the $TM_{total}$ of the core-only control volume. In other words, the header is critical to the adsorber $TM_{total}$ and should not be neglected.

**Table 6.** Comparison of thermal mass in header with HX core.

| Component (Section) | Header Mass (Mass Difference between HX-Only and HX Core-Only) | | Header Thermal Mass (Difference between HX-Only and HX Core-Only) | Thermal Mass, HX Core-Only |
|---|---|---|---|---|
| | HTF (kg) | Metal (kg) | $(kJ^1K^{-1})$ | $(kJ^1K^{-1})$ |
| Adsorber, flat tube–fin, coated (Section 3.4) | 0.15 | 0.13 | 0.76 | 1.21 |
| Adsorber, round tube–fin, packed (Section 3.5) [1] HTF: silicon oil | 0.30 | 1.49 | 1.08 | 1.90 (0.5 kg packed) 2.12 (0.766 kg coated) |
| Adsorber, round tube–fin, packed (Section 3.5) [1] HTF: water | 0.33 | 1.49 | 1.93 | 3.22 (0.5 kg packed) 3.44 (0.766 kg coated) |
| Adsorber, shell–tube (Section 3.7) | 0.1 | 0.25 | 0.52 | 2.23 |
| Adsorber, fiber heat exchanger (Section 3.9) | 0.83 | 2.23 | 5.48 | 13.08 |

[1] Values in this row are estimates since exact geometrical information on the header was not available.

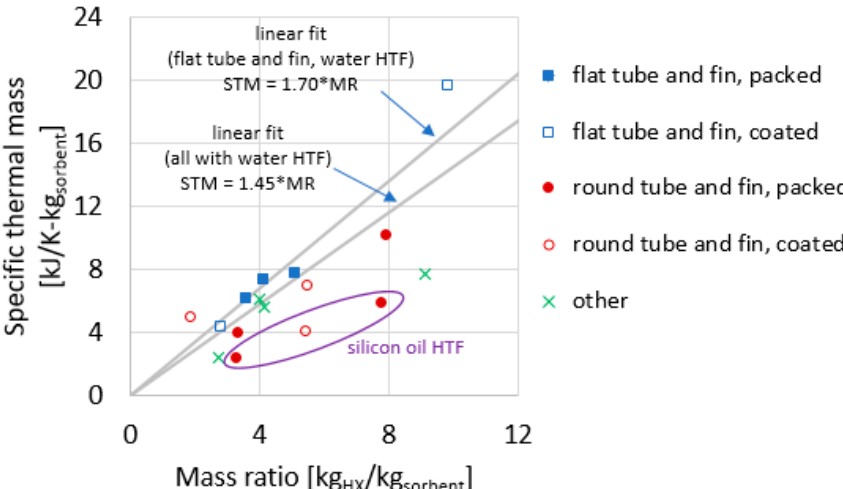

**Figure 12.** Specific thermal mass plotted against the mass ratio for the heat exchangers reported in this study.

Future work should consider the effect of additional thermal masses that are included in the "comprehensive" control volume.

To further illustrate the importance of HTF to $TM_{total}$ and $STM$, consider the case of a single round tube–fin packed adsorber (Section 3.5) that was used with two different HTFs: water and silicon oil. The silicon oil had a slightly lower density (910 vs. 1000 $kg^1 m^{-3}$) and a much lower specific heat (1.73 vs. 4.19 $kJ^1 kg^{-1} K^{-1}$). As a result, referring to Table 4, the $TM_{total}$ reduced from 5.15 to 2.98 $kJ^1 K^{-1}$ (42% lower), with a corresponding drop in $STM$ from 10.30 to 5.96 $kJ^1 K^{-1} kg_{sorb}^{-1}$ (also 42%). This occurred with only a minor change in $MR$, from 7.87 to 7.72 $kg_{HX}^1 kg_{sorb}^{-1}$ (2% lower).

## 5. Conclusions

$TM_{total}$ and $STM$ are useful to the analysis of sorption systems by providing measures of the thermal mass of sorption components. New experimental data for the thermal masses of nine sorption HXs are presented. The thermal mass of the shell or enclosure was not included. The overall $STM$ varied from 2.46 to 19.51 $kJ^1 K^{-1} kg_{sorb}^{-1}$.

The reported $c_{effective}$ ranged from 0.77 to 2.00 $kJ^1 K^{-1} kg_{HX}^{-1}$. Because of this wide range of values, the $STM$ is not generally accurately predictable for a generic adsorption HX. A $c_{effective}$ of 1.45 can serve as a general rule of thumb for attaining the $TM_{total}$ for a flat tube–fin adsorber from the readily calculated $MR$.

However, the accuracy of $c_{effective}$ is much better when consideration is restricted to flat tube–fin type geometry (either packed or coated adsorbent) with water as HTF. With these constraints, the $STM$ was predicted with an RMSE of 9% and a worst-case prediction error of 15% by multiplying the simple $MR$ ($m_{HX}/m_{sorb}$) by an empirically fitted $c_{effective}$ of 1.70 $kJ^1 K^{-1} kg_{HX}^{-1}$.

This rule of thumb can be useful for estimating the $TM_{total}$ of a flat tube–fin HX when only its $MR$ is readily obtainable. As more data are compiled in the literature, it may become possible to develop additional guidelines for $c_{effective}$ for different HX types.

**Author Contributions:** Conceptualization, K.R.G.; Data curation, K.R.G.; Investigation, A.F., A.V., S.M., Z.Y., M.R., and C.B.; Methodology, K.R.G., A.V., S.M., and M.R.; Visualization, K.R.G.; Writing—original draft, K.R.G., A.F., A.V., S.M., Z.Y., and M.R.; and Writing—review and editing, K.R.G., A.F., A.V., S.M., Z.Y., M.R., C.B., M.Q., E.L., A.R.-P., S.H., R.C., M.B., G.F. and I.H. All authors have read and agreed to the published version of the manuscript.

**Funding:** This work was partially sponsored by the US Department of Energy's Building Technologies Office under Contract No. DE-AC05-00OR22725 with UT-Battelle, LLC. The authors would like to acknowledge Mr. Antonio Bouza, Technology Manager HVAC&R, Water Heating, and Appliance, US Department of Energy Building Technologies Office. Funding of the German Ministry of Economics and Energy within the project "ADOSO" (FKZ

03ET1127) for the work on the fiber adsorber is gratefully acknowledged. The work on binder coated adsorbers was supported by the Fraunhofer Zukunftsstiftung under grant HARVEST and by the German Federal Ministry of Education and Research (BMBF) under grant 03SF0441B. Funding is acknowledged from UK EPSRC Grant EP/K011847/1 i-STUTE. Funding is acknowledged from Natural Sciences and Engineering Research Council of Canada (NSERC) through the Automotive Partnership Canada Grant No. APCPJ 401826-10.

**Acknowledgments:** The authors would like to acknowledge the IEA Heat Pump Program Annex 43 "Fuel Driven Sorption Heat Pumps" for motivation and establishing working relationships among institutions, Kathy Jones for formatting, and Olivia Shafer for technical editing.

**Conflicts of Interest:** The authors declare no conflict of interest.

## Nomenclature

| | |
|---|---|
| A | area (m$^2$) |
| AC | activated carbon |
| AT | approach temperature (K) |
| c | specific heat capacity (kJ$^1$kg$^{-1}$K$^{-1}$) |
| $c_{effective}$ | effective specific heat |
| HTS | high temperature salt |
| HX | heat exchanger |
| LTS | low temperature salt |
| m | mass (kg) |
| *MR* | mass ratio (kg$_{HX}$/kg$_{sorbent}$) |
| *STM* | specific thermal mass |
| *TM* | thermal mass (kJ/kg) |
| V | volume (m$^3$) |
| Y | ratio of refrigerant (sorbate) mass to sorbent mass (kg$_{ref}$/kg$_{sorbent}$) |
| Greek | |
| $\rho$ | density (kg/m$^3$) |
| $\omega$ | humidity ratio (kg$_w$/kg$_{da}$) |
| Subscripts | |
| 0 | initial |
| HTF | heat transfer fluid |
| liq | liquid |
| mat | material (non-HTF, non-sorbent), typically metals, comprising the heat exchanger |
| total | total (thermal mass) |
| ref | refrigerant (sorbate) |
| sorb | sorbent |

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
