# Peer review of "Experimentally Measured Thermal Masses of Adsorption Heat Exchangers"

_energies, doi:10.3390/en13051150_

Round 1

Reviewer 1 Report

This manuscript provides sufficient novel scientific information. No correction required from my end. It can be accepted in the current form.

Reviewer 2 Report

Please avoid lumping references. It makes the state of the art inappropriate.

Double-check typo errors and grammar mistakes.

Reviewer 3 Report

As attached. 

Reviewer 4 Report

General overview:

This work deals with developing a methodology based on experimental data extracted from literature to take into account the impact of the components' thermal mass of a system. The study focuses particularly on different kinds of heat exchanger devices and materials as well the heat transfer fluids.

The context is well set and documented in the introduction.

The trends obtained cannot be reliable with so few available experimental points (Cf. Figures 11 and 12) but the proposed methodology is astute and can be very useful in a further and thougher work.

Minor corrections:

Table 2 : the acronyms of HTS and LTS are missing.

l 47: "quantity" instead of "quanity".
